# Plaque Reduction Neutralization Test (PRNT) in the Congenital Zika Syndrome: Positivity and Associations with Laboratory, Clinical, and Imaging Characteristics

**DOI:** 10.3390/v12111244

**Published:** 2020-10-31

**Authors:** Marizélia R. C. Ribeiro, Ricardo Khouri, Patrícia S. Sousa, Maria R. F. C. Branco, Rosângela F. L. Batista, Elaine P. F. Costa, Maria T. S. S. B. Alves, Gláucio A. Amaral, Marcella C. R. Borges, Eliana H. M. Takahasi, Líllian N. Gomes, Ana K. T. Mendes, Tamires B. Cavalcante, Luciana C. Costa, Alvina C. Félix, Nathalia C. S. Souza, Antônio A. M. Silva

**Affiliations:** 1Department of Medicine III, Federal University of Maranhão, Praça Gonçalves Dias, 21, Centro, São Luís CEP 65020-240, MA, Brazil; 2Laboratory of Vector-Borne Infectious Diseases, Gonçalo Moniz Institute, Fiocruz-Bahia, Rua Waldemar Falcão, 121, Candeal, Salvador CEP 40296-710, BA, Brazil; ricardo_khouri@hotmail.com; 3Department of Pathology and Legal Medicine, Faculty of Medicine, Federal University of Bahia, Praça XV de Novembro, s/n–Largo do Terreiro de Jesus, Salvador CEP 40026-010, BA, Brazil; 4Reference Center on Neurodevelopment, Assistance and Rehabilitation of Children/NINAR– Health Secretariat of the State of Maranhão, Av. dos Holandeses, s/n, São Luís CEP 65077-357, MA, Brazil; cdneuropatricia@gmail.com; 5Department of Pathology, Federal University of Maranhão, Praça Madre Deus, 2, Madre Deus, São Luís CEP 65025-560, MA, Brazil; mrfcbranco@gmail.com; 6Department of Public Health, Federal University of Maranhão, Rua Barão de Itapary, 155, Centro, São Luís CEP 65020-070, MA, Brazil; rosangela.flb@ufma.br (R.F.L.B.); mtssb.alves@ufma.br (M.T.S.S.B.A.); anakaroltm@gmail.com (A.K.T.M.); tamiresbarradas@gmail.com (T.B.C.); luciana13cavalcante@gmail.com (L.C.C.); silva.antonio@ufma.br (A.A.M.S.); 7Department of Medicine I, Federal University of Maranhão, Praça Gonçalves Dias, 21, Centro, São Luís CEP 65020-240, MA, Brazil; elaine.pfc68@gmail.com; 8Sarah Network of Neurorehabilitation Hospitals, Av. Luis Rocha, s/n, Monte Castelo, São Luís CP 65035-270, MA, Brazil; glaucio_amaral@yahoo.com.br (G.A.A.); elimorioka@gmail.com (E.H.M.T.); 9Diagnostic Imaging Department, Federal University of São Paulo, Rua Napoleão de Barros, 800, São Paulo CEP 04024-002, SP, Brazil; mborges@huhsp.org.br; 10Laboratory of Human Immunology, Department of Immunology, Institute of Biomedical Sciences, University of São Paulo, Av. Prof. Lineu Prestes, 1730, Cidade Universitária, São Paulo CEP 05508-900, SP, Brazil; lilliangomes20@gmail.com; 11Laboratory of Virology, Institute of Tropical Medicine, University of São Paulo, Av. Dr. Enéas Carvalho de Aguiar, 470, São Paulo CEP 05403-000, SP, Brazil; clarafelix@usp.br (A.C.F.); nathalia_santiago21@hotmail.com (N.C.S.S.)

**Keywords:** arbovirus, birth defects, Congenital Zika Syndrome, infant health, pregnancy

## Abstract

The short duration of viremia, low blood viral load, inaccessibility to timely specific diagnostic tests, and cross-reactions with other flaviviruses have hindered laboratory confirmation of Congenital Zika Syndrome (CZS). This study analyzes the positivity of the plaque reduction neutralization test (PRNT) in children with clinical or imaging characteristics of CZS and its association with laboratory, clinical, and imaging characteristics. The 94 clinical cases of CZS submitted to the ZIKV PRNT_90_ test were followed from 2016 to 2018. The mean age of children at PRNT_90_ collection was 22 ± 6 months Standard Deviation. The ZIKV PRNT_90_ was positive (titer ≥ 10) in 40 (42.5%) children. ZIKV PRNT_90_ positivity was associated with severe microcephaly in newborns (*p* = 0.016), lower head circumference z-score at birth (*p* = 0.043) and 24 months of age (*p* = 0.031), and severe reduction of the cerebral parenchyma volume (*p* = 0.021), expressing greater disease severity. Negative PRNT_90_ in children with characteristic signs of CZS may be due to false-negative results, indicating that the diagnosis of CZS should be primarily syndromic.

## 1. Introduction 

The confirmation of Congenital Zika Syndrome (CZS) by specific laboratory tests became a challenge when the microcephaly epidemic started in Brazil in 2015 [1] and spread to other locations in Latin America [2,3]. The short duration of viremia [4,5,6], low blood viral loads [6], cross-reactions with other flaviviruses in serological tests [5,6], and difficulties in accessing specific and accurate diagnostic tests [1,5,7] have delayed or hampered laboratory confirmation of typical CZS clinical cases [1,2,3]. 

A positive ZIKV ribonucleic acid test (RNA NAT) in serum [5,7], urine [5], or cerebrospinal fluid (CSF) confirms CZS. A positive ZIKV IgM [5,7] in cases in which ZIKV RNA NAT was negative or was not performed should be considered a likely CZS case due to a possible cross-reaction with other flaviviruses (false-positive result). In this situation, a plaque reduction neutralization test (PRNT) positive for ZIKV and negative for dengue virus (DENV) is the confirmation criterion for CZS [5,8]. The CDC recommends that this test be run after 18 months of age when maternal antibodies are no longer present in the child [5]. If both ZIKV RNA NAT and ZIKV IgM are negative, the child is unlikely to have CZS [5,8,9]. 

Difficulties in carrying out tests or inconclusive test results meant that most children born with congenital disabilities during and after the 2015 microcephaly outbreak were diagnosed with likely CZS [3,10,11,12,13,14,15,16,17,18]. A positive ZIKV IgM antibody by ELISA was the most widely used evidence for the presumed or confirmed diagnosis of CZS in several case series [10,12,13,15,18,19,20,21,22,23,24,25,26]. A negative ZIKV RT-PCR was found in some children with typical signs of CZS [13,15,17,18,19,21,22,25,27]. The ZIKV PRNT, a high-cost and time-consuming test [9], was hardly used for diagnosing CZS [20,24,26,28].

Although the CDC indicated ZIKV PRNT as a diagnostic criterion to confirm ZIKV infection and rule out cross-reactions (false-positive results) with DENV, three studies reported false negative results for this test. A study in a cohort of pregnant women with rash found low positivity (48.5%) of ZIKV PRNT in positive ZIKV RT-PCR pregnant women [29]. In a case-control study which included 91 cases of CZS with typical microcephaly (82 live births and nine stillbirths), 27 mothers were PRNT_50_-negative [22]. A study with 19 confirmed, possible or likely cases of CZS revealed negative PRNT in five normocephalic newborns, but with changes in cranial tomography suggestive of CZS. Two of these newborns were born to ZIKV PRNT-positive mothers during pregnancy and had negative results for ZIKV IgM, syphilis, toxoplasmosis, cytomegalovirus, and rubella [20].

Based on possible false-negative PRNT results in children with likely CZS [20] and pregnant women with positive ZIKV RT-PCR [22,29], this study hypothesized that there are few differences in laboratory, clinical, and imaging characteristics comparing positive and negative PRNT children with typical CZS. 

## 2. Materials and Methods

### 2.1. Type of Study and Data Collection

This cohort study collected data from 134 children referred to the Reference Center for Neurodevelopment, Care, and Rehabilitation of Children (NINAR/Maranhão State Government/Brazil) to investigate the congenital Zika syndrome CZS). The children were born between October 2015 and September 2018 in different municipalities in the state of Maranhão.

Maranhão is a state in the Brazilian Northeast of 217 municipalities, with an estimated 6,904,298 inhabitants for 2015 [30]. In 2016, at the peak of the epidemic, Maranhão had a Municipal Human Development Index (MHDI) of 0.682, the second worst in Brazil [31], and a Gini Index of 0.528 [32].

### 2.2. Case Definition

These 134 children were divided into five groups based on the classification by França et al. [11]. Likely laboratory cases had a positive M-class specific immunoglobulin (IgM) serological test for the Zika virus. Highly likely clinical cases had cranial tomographic changes suggestive of CZS (calcifications, reduced cerebral parenchyma volume, ventriculomegaly, malformation of the cortical development, cerebellum malformation/hypoplasia, brain stem malformation/hypoplasia, and corpus callosum agenesis/dysgenesis) and negative serologies for toxoplasmosis, cytomegalovirus, and syphilis. Moderately likely clinical cases had cranial tomographic changes suggestive of CZS and one or more serologies for toxoplasmosis, cytomegalovirus, and syphilis that were not performed or were inconclusive. In the unlikely cases, changes in imaging studies were not described in detail, and the results for syphilis, toxoplasmosis, and cytomegalovirus were negative or not performed. The cases not included in these four groups were not considered to be Zika cases.

This study’s inclusion criterion was having undergone the PRNT_90_ test and belonging to one of the following groups: Likely laboratory cases, highly likely clinical cases, and moderately likely clinical cases. Exclusion criteria were not having performed the PRNT_90_ test or having a negative ZIKV PRNT_90_ test and belonging to groups of unlikely cases or that are not CZS cases.

### 2.3. Plaque Reduction Neutralization Test (PRNT90)

The 90% plaque reduction neutralization test (PRNT90) was performed in the Laboratory of Vector-borne Infectious Diseases (LEITV)–Gonçalo Moniz Institute/Oswaldo Cruz Foundation (FIOCRUZ)/Bahia for the laboratory confirmation of CZS cases. PRNT was performed based on a previously reported protocol with minor modifications. The cutoff value for PRNT positivity was defined as 90% (PRNT_90_). PRNT_90_ was performed to determine the maximum serum dilution (1:8 to 1:1024) needed to reduce arbovirus plaque formation by 90% among Vero cells. Thus, the ZIKV virus strain (ZIKV/H.sapiens/Brazil/PE243/2015-Asian, Recife, Brazil) was used. All sera were heat-inactivated (56 °C, 30 min) before neutralization testing. The serum samples were diluted with the serial dilutions method, using a modified Dulbecco Eagle medium containing 2% fetal calf serum and 1% of Penicillin/Streptomycin as diluent. Next, virus suspension was mixed to each serum dilution and incubated at 37 °C for 60 min. A final volume of each serum dilution and virus (50–100 ffu/well–12-well cell culture plate) mixture was transferred to a well containing Vero cells and then incubated at 37 °C for 60 min. Following incubation, 0.3% agarose solution was added, and plates were re-incubated at 37 °C for 5 days. Reactions were then revealed using a 2% naphthol blue-black solution. Titers ≥ 10 were considered positive [33].

### 2.4. Anti-DENV and Anti-ZIKV IgG Antibodies

All additional serologic tests (commercial ELISA) were performed by the Laboratory of Virology from the Institute of Tropical Medicine–University of São Paulo (IMT-USP). The samples were children’s serum samples previously tested in the PRNT assay and serum samples from their mothers collected in the same period. Anti-DENV and anti-ZIKV IgG antibodies were detected using, respectively, Dengue Virus IgG DxSelect™ (FOCUS Diagnostics, Cypress, USA) and Anti-Zika Virus ELISA (IgG) (Euroimmun, Lübek, Germany) ELISA commercial kits. All tests were performed according to the manufacturer’s instructions. At the first consultation at NINAR, five children had collected blood for ZIKV RT-PCR and 27 for M-class immunoglobulin (IgM). These data were taken from medical records.

### 2.5. Clinical and Imaging Variables

Variables that represented clinical and imaging characteristics were collected from medical records, interviews with the children’s mothers or guardians, reports, medical reports, and pregnant women’s and children’s cards. The RT-PCR results of five mothers and children were obtained in interviews, with verification of test results.

The head circumference (HC) was measured by the neuropediatrician of the research group and by trained health professionals. The calculation of the HC’s Z-score at birth followed the INTERGROWTH-21^st^ standards for gender and gestational age, using adjusted gestational age for preterm children [34]. The calculation of HC’s Z-score at 6, 12, 24, and 36 months followed the World Health Organization’s standard curve for head circumference [35]. Mild microcephaly was defined as HC < 2 and ≥ 3 standard deviations (SD) below the mean for gestational age and gender, and severe microcephaly when HC < 3 standard deviations below the mean [7].

The variables that measured maternal characteristics were rash during pregnancy and symptoms of Zika virus infection in the first gestational semester, dichotomized into NO and YES. The newborn’s clinical characteristics were microcephaly at birth (categorized as no microcephaly, microcephaly, and severe microcephaly) and the HC’s z-score at birth, analyzed as a continuous variable.

The clinical variables in children aged 2 to 36 months were as follows: (a) Variables that represented the CZS clinical phenotype (craniofacial disproportion, biparietal depression, occipital protuberance, frontotemporal retraction, excess nuchal skin, and suture stripping, dichotomized into NO and YES); (b) ophthalmological abnormalities (mobilization of macular pigment and chorioretinal scar, dichotomized into NO and YES); (c) neurological abnormalities (age of first seizure and drug-resistant epilepsy); and (d) assessment of motor development. The age at first seizure was classified into no seizure, 0–5 months, 6–12 months, and 12 months and over. Drug-resistant epilepsy, defined as epilepsy refractory to two anticonvulsant medications [36], was dichotomized into NO and YES.

Motor development was classified according to the Gross Motor Function Classification System (GMFCS, School of Rehabilitation Science, Ontario, Canada), which describes five gross motor function patterns in children with cerebral palsy. Level I children walk without limitations; level II children walk with limitations; those included in level III walk with assistance; those in level IV have wheelchair automobility; and those in group V are transported in a wheelchair. In this study, children were divided into groups with mild/moderate impairment (levels I, II, and III) and with severe impairment (levels IV and V) [37].

The infants underwent a full eye examination, performed by the same ophthalmologist every three months in the first year of life, which included anterior segment evaluation, motility evaluation, retinoscopy, binocular indirect ophthalmoscopy, and wide-angle fundus photography (RetCam Shuttle; Clarity Medical Systems, Pleasanton, USA), performed with mydriasis (tropicamide 1%).

The variables that assessed cranial tomography characteristics were reduced volume of cerebral parenchyma (no reduction, mild/moderate/severe), brain calcifications (no calcifications, subcortical, and other locations), degree of ventriculomegaly (no ventriculomegaly, mild/moderate, and severe), type of ventriculomegaly (no ventriculomegaly, ex vacuo, and hypertensive), malformation of cortical development (NO and YES), cerebellum malformation (NO and YES), and brainstem malformation (NO and YES).

Brain computed tomography through volumetric acquisitions in the Siemens AS Definition multi-slice devices (64 channels, Siemens, München Germany) was performed to identify cranial abnormalities. Images were analyzed individually by two experienced neuroimaging radiologists. After their assessments, they discussed divergent results and reached a consensus.

### 2.6. Statistical Analysis

All information obtained in the CZS NINAR cohort was stored in a REDCap (Vanderbilt University, Nashville, USA) software database and transferred to the Stata^®^ 14.0 program for statistical analysis.

The categorical variables were presented as relative and absolute frequencies and continuous variables as medians and interquartile range (IQR). The chi-square test or Fisher’s exact test was used to verify the association between categorical variables and the T-test to compare the HC’s z-score means.

### 2.7. Ethics Statement

Written informed consent was obtained from all children’s guardians to acquire this data. The Institutional Research Ethics Committee of the University Hospital of the Federal University of Maranhão approved the study on 6 September 2017 under opinion number (CAAE) 65897317.1.0000.5086.

## 3. Results

Forty negative ZIKV PRNT_90_ children from the groups of unlikely CZS cases or no CZS were excluded from the total of 134 children referred to NINAR for investigation of CZS. Ninety-four children were included in this study, of which 40 (42.5%) had laboratory confirmation of CZS by the ZIKV-specific PRNT_90_ and 54 (57.5%) had negative ZIKV PRNT_90_ from the laboratory likely cases groups (3 children; 3.2%), highly likely clinical cases (26 children; 27.7%), and moderately likely clinical cases (25 children; 26.6%).

The three likely laboratory cases (positive ZIKV IgM and negative PRNT_90_) had a severe reduction in the cerebral parenchyma volume, severe symmetrical ventriculomegaly (in two ex vacuo and one hypertensive), and malformations of the cortical development. In two of these children, brain calcifications predominated in the cortico-subcortical region. Brain calcifications were challenging to characterize in the third child due to severe cerebral atrophy. Two children had cerebellum hypoplasia/malformation, and one of them also had brain stem hypoplasia/malformation. The mothers of the three children reported symptoms suggestive of ZIKV infection during pregnancy. These children were 22, 24, and 25 months old when they collected blood for the PRNT_90_ (results not shown in the table).

The ages of the 94 children at the first consultation at NINAR ranged from less than 1 month to 31 months, with a median of 7 months. Children were 2 to 35 months old (mean 22 ± 6 months standard deviation) at the PRNT^90^ collection. Five children had undergone ZIKV RT-PCR with negative results before the referral to NINAR. Two of them had positive ZIKV PRNT_90_, and one also had positive ZIKV IgM. The serum collections of these two children for ZIKV RT-PCR and IgM were performed on the first day of life (results not shown in the table).

Of a total of 88 children who underwent IgG Euroimmun for the Zika virus, only six (6.8%) children were positive for this test and two of them were ZIKV PRNT_90_-positive (Table 1). These six children were over 18 months of age at the test collection (result not shown in tables). Ninety mothers underwent IgG Euroimmun for ZIKV, and 78 (86.7%) were positive. IgG Focus for the Dengue virus positivity was superior to the IgG Euroimmun for the Zika virus, both in children (46.6%) and in mothers (97.8%). There were no differences between ZIKV PRNT_90_ positivity, and ZIKV IgM results in children, and IgG for Zika and dengue virus in children and mothers Table 1. All ZIKV IgG positive children and mothers were DENV IgG-positive (result not shown in table).

There were o differences between the presence of maternal symptoms during pregnancy and a positive ZIKV PRNT_90_ (*p* > 0.05) Table 2.

Regarding the children’s clinical characteristics, PRNT_90_ positivity was more frequent when newborns had severe microcephaly (*p* = 0.016; Table 3) and lower head circumference z-score at birth (*p* = 0.043) and 24 months of age (*p* = 0.031; Table 4).

Regarding cranial tomography results in children, we observed an association between positive ZIKV PRNT_90_ and severe reduction in cerebral parenchyma volume (*p* = 0.021; Table 5).

## 4. Discussion

ZIKV PRNT_90_ positivity was 42.5% in the CZS NINAR cohort. Severe microcephaly in newborns, lower head circumference z-scores at birth and 24 months of age, and severe reduction in the cerebral parenchyma volume were the clinical and imaging characteristics of children associated with a positive ZIKV PRNT_90_.

This positivity was less than expected by the authors, considering that this test was negative in three likely laboratory cases (3.2% of negative PRNT_90_) and 26 highly likely clinical cases (26.7% of negative PRNT_90_). This low positivity cannot be attributed to problems in running the PRNT_90_, because the protocol established by Baer and Kehn-Hall [33] was strictly followed, and the strain ZIKV/H.sapiens/Brazil/PE243/2015-Asian identified in the State of Pernambuco at the time of the CZS epidemic in Brazil [25] was used.

Given that 57.5% negative ZIKV PRNT_90_ in the CZS NINAR cohort were primarily false-negatives, we highlight ZIKV PRNT-negative findings in four studies. Ximenes et al. found 51.5% of ZIKV PRNT_50_ negativity in a cohort of 103 pregnant women who reported rash during pregnancy and were ZIKV RT-PCR-positive. The RT-PCR was collected on the same day of onset or up to 72 h after the start of rash, and the PRNT_50_ was performed within 1 year of the rash [29]. In the study by Araújo et al., CSF-positive ZIKV IgM newborns were born to PRNT_50_-negative mothers [22]. Working with data from this latest study, Castanha et al. found higher ZIKV PRNT levels in newborns than in their respective mothers (*p* < 0.001) [38]. Finally, Aragão et al. underscored the fact that two normocephalic newborns born to positive ZIKV PRNT mothers had negative ZIKV IgM in the CSF and PRNT. These children also had negative results for syphilis, toxoplasmosis, cytomegalovirus, rubella, and dengue in tests collected shortly after birth. Cranial tomography performed when these two children were 11 months old showed cortico-subcortical white matter junction calcifications [20], one of the five characteristics that differentiate CZS from other congenital infections [39,40].

To understand possible false-negative PRNT results, we evidenced that only six children (two positive PRNT_90_ and four negative PRNT_90_) from the ZIKA NINAR cohort had positive ZIKV IgG out of a total of 88 samples tested for this method. ZIKV IgG Euroimmun positivity cannot be attributed to maternal transfer, because these six children were over 18 months old when they collected blood, as indicated by the CDC [5]. Two of these children had higher immunoglobulin titers than their mothers.

Given that most children in the ZIKA NINAR cohort did not produce ZIKV IgG, a case report in which ZIKV IgG was measured at birth and 9, 12, 21, and 24 months using indirect immunofluorescence tests based on ZIKA virus-infected cells as the antigenic substrate contributed to explaining the high negativity of ZIKV IgG in our study. ZIKV IgG was positive at birth and when the child was 12 months of age and was negative at 21 and 24 months of age. PRNT was only collected at birth and was positive. The child’s mother had positive ZIKV RT-PCR, IgM, IgG, and PRNT when pregnant. The study authors raised three hypotheses for this negative IgG finding: (a) A low and transient ZIKV viremia can lead to a lack of antigen detection and the absence of a strong immune response by the child; (b) a specific and direct ZIKV tropism to the central nervous system can cause ZIKV to become a silent virus that escapes the host’s response; and (c) ZIKV could prevent the triggering of a strong innate immune response because active viral replication may have ceased during intrauterine life or shortly after that [41].

The low anti-ZIKV IgG seropositivity in newborn find in this study might also be explained by the use of recombinant NS1 antigen as target in the commercial kit. A lower antibody affinity maturation against ZIKV nonstructural protein NS1 than structural envelop (E) protein following Zika virus infection in adults has been demonstrated [42]. In addition, serum-neutralizing activity appears to correlate with levels of antibodies to the ZIKV E protein domain III (ZEDIII) [43]. Furthermore, ZEDIII antibodies are significantly increased in mother’s serum of microcephalic newborns, while NS1 are significantly decreased [44].

In the CZS NINAR cohort, we observed a tendency for positive ZIKV PRNT_90_ children to have CZS with greater severity concerning brain impairment. These children were more often born with severe microcephaly and had lower head circumference z-scores at birth and 24 months of age, as well as severe reduction in the cerebral parenchyma volume on cranial tomography. In the study by Aragão et al., the five children born normocephalic with negative ZIKV PRNT had less brain impairment on imaging tests than those born with microcephaly [20].

This study has limitations regarding the laboratory confirmation of CZS before the PRNT_90_ results. Only five children had undergone ZIKV-specific RT-PCR before the first visit at NINAR, and all results were negative, including those of the three children who had this test collected timely, as determined by the CDC [5]. Until 2017, the ZIKV RT-PCR was not performed in public laboratories in the state of Maranhão [1], the mothers’ place of residence during pregnancy, and for all deliveries in this series of cases, hindering laboratory confirmation of CZS. ZIKV IgM was performed on only 27 children, with 17 negative results. However, negative ZIKV RT-PCR and IgM should not exclude the diagnosis of CZS [5,45]. Another limitation of the study is selection bias, because the sample included mostly children with severe cerebral palsy.

This study has three strengths. The first is the use of a strain isolated in Recife (Pernambuco), where the microcephaly outbreak began in Brazil in 2015, for the PRNT_90_. Therefore, it is unlikely that the test’s negativity was due to a mutation of the strain. The second strength is that the PRNT_90_ was performed at an age when most children should no longer have maternal antibodies [5], and we were unable to identify another study testing the positivity of PRNT in children in the same age group [20,22,24,26]. Finally, the sample size, which allowed us to detect some differences between the positive and negative PRNT groups, is also a strength of this study.

## 5. Conclusions

In the NINAR cohort of CZS, positive and negative ZIKV PRNT_90_ children showed few laboratory, clinical, and cranial tomography differences. This negligible difference between the positive and negative PRNT_90_ groups suggests that PRNT_90_ may be false-negative in CZS cases. The findings point out that the diagnosis of CZS should be predominantly syndromic if the test collection exceeds the period of qRT-PCR positivity.

## Figures and Tables

**Table 1 viruses-12-01244-t001:** Laboratory tests in children and mothers and positive and negative plaque reduction neutralization test in children. São Luís, Brazil, 2016–2018.

Laboratory Characteristics	PRNT_90_ ^a^ Positive ^b^		PRNT_90_ ^a^ Negative ^c^		*p*-Value
	f	%	f	%	
In children					
ZIKV IgM ^d,h^ (*n* = 27)					
Negative	7	41.2	10	58.8	0.236
Positive	7	70.0	3	30.0	
Age (in months) of child at collection of PRNT90 ^a^ (*n* = 88)					0.280
<12	3	60.0	2	40.0	
12 to 17	2	20.0	8	80.0	
≥18	35	44.3	44	55.7	
ZIKV IgG ^e,h^ (*n* = 88)					0.219
Negative	30	39.0	47	61.0	
Positive ^g^	2	33.3	4	66.7	
Inconclusive	4	80.0	1	20.0	
Dengue IgG ^f,i^ (*n* = 88)					0.195
Negative	16	34.0	31	66.0	
Positive	20	48.8	21	51.2	
In mothers					
ZIKV IgG ^e,h^ (*n* = 90)					0.447
Negative	5	62.5	3	37.5	
Positive ^g^	32	41.0	46	59.0	
Inconclusive	1	25.0	3	75.0	
Dengue IgG ^f,h^ (*n* = 90)					0.669
Negative	1	50.0	1	50.0	
Positive	37	42.0	51	58.0	

^a^ 90% Plaque Reduction Neutralization Test (PRNT). ^b^ Titer ≥ 10. ^c^ Titer < 10. ^d^ M-class specific immunoglobulin (IgM). Results from medical notes. ^e^ Euroimmun Anti-Zika Virus ELISA class Immunoglobulin (IgG). ^f^ FOCUS IgG Class Immunoglobulin. ^g^ All children were also dengue-specific IgG-positive. ^h^ Fisher’s exact test. ^i^ Chi-square test.

**Table 2 viruses-12-01244-t002:** Symptoms of infection in pregnant women and positive and negative plaque reduction neutralization test in children with Congenital Zika Virus (CZS). São Luís, Brazil, 2016–2018.

Pregnant Women’s Infection Symptoms	PRNT90 ^a^ Positive ^b^		PRNT90 ^a^ Negative ^c^		*p*-Value
	f	%	f	%	
Rash during pregnancy ^d^ (*n* = 90)					0.424
No	17	48.6	18	51.4	
Yes	22	40.0	33	60.0	
Presence of Zika virus infection symptoms during pregnancy ^d^ (*n* = 90)					0.637
No	9	39.1	14	60.9	
Yes	30	44.8	37	55.2	
Zika virus infections symptoms in the first trimester of pregnancy ^d^ (*n* = 67)					0.981
No	9	45.0	11	55.0	
Yes	21	44.7	26	55.3	

^a^ 90% Plaque Reduction Neutralization Test. ^b^ Titer ≥ 10. ^c^ Titer < 10. ^d^ Chi-square test.

**Table 3 viruses-12-01244-t003:** Clinical and cranial tomography characteristics of children with CZS and positive and negative plaque reduction neutralization test. São Luís, Brazil, 2016–2018.

Clinical Characteristics	PRNT90 ^a^ Positive ^b^		PRNT90 ^a^ Negative ^c^		*p*-Value
	f	%	F	%	
Microcephaly at birth ^d^ (*n* = 79)					0.016
No microcephaly	8	26.7	22	73.3	
Microcephaly	4	28.6	10	71.4	
Severe microcephaly	21	60.0	14	40.0	
Congenital Zika Syndrome clinical phenotype (*n* = 93)					
Craniofacial disproportion ^d^					0.523
No	3	37.5	5	62.5	
Yes	37	43.5	48	56.5	
Biparietal depression ^e^					0.447
No	21	39.6	32	60.4	
Yes	19	47.5	21	52.5	
Occipital protuberance					0.295
No	16	37.2	27	62.8	
Yes	24	48.0	26	52.0	
Fronto-temporal retraction ^e^					0.457
No	10	37.0	17	63.0	
Yes	30	45.5	36	54.5	
Excess nuchal skin ^e^					0.476
No	30	42.1	43	58.9	
Yes	10	50.0	10	50.0	
Suture stripping ^e^					0.623
No	26	41.3	37	58.7	
Yes	14	46.7	16	53.3	
Ophthalmic changes (*n* = 87)					
Mobilization of macular pigment ^e^					0.955
No	30	46.2	35	53.8	
Yes	10	45.5	12	54.5	
Chorioretinal scar ^e^					0.891
No	31	45.6	37	54.4	
Yes	9	47.4	10	52.6	
Neurological changes					
Age of first seizure ^d^ (*n* = 91)					0.387
No seizure	6	66.7	3	33.3	
0-5 months	19	39.6	29	60.4	
6-11 months	8	38.1	13	61.9	
>12 months	7	53.8	6	46.2	
Drug-resistant epilepsy (*n* = 91) ^e^					0.290
No	19	50.0	19	50.0	
Yes	21	38.9	33	61.1	
Assessment of motor development with GMFCS ^d,f^ (*n* = 89)					0.200
I to III	1	16.7	5	83.3	
IV and V	36	56.6	47	43.4	

^a^ Plaque Reduction Neutralization Test at 90%. ^b^ Titer ≥ 10. ^c^ Titer < 10. ^d^ Fisher’s exact test. ^e^ Chi-square test. ^f^ Gross Motor Function Classification System (GMFCS).

**Table 4 viruses-12-01244-t004:** Means of the Z scores of the head circumference at birth and at 6, 12, 24, and 36 months and positive and negative plaque reduction neutralization test in cases of CZS. São Luís, Brazil, 2016–2018.

Z-score of Head Circumference ^d^	PRNT90 ^a^ Positive ^b^		PRNT90 ^a^ Negative ^c^		*p*-Value
	N	Mean	N	Mean	
At birth	33	−3.15	46	−2.43	0.043
6 months	24	−6.10	24	−5.16	0.169
12 months	28	−5.86	37	−4.98	0.113
24 months	33	−5.71	45	−4.77	0.031
36 months	15	−6.17	22	−4.72	0.061

^a^ 90% Plaque Reduction Neutralization Test. ^b^ Titer ≥ 10. ^c^ Titer < 10. ^d^ T-test of independent samples.

**Table 5 viruses-12-01244-t005:** Changes in cranial tomography and positive and negative plaque reduction neutralization test in children with CZS. São Luís, Brazil, 2016–2018.

Changes in Cranial Tomography	PRNT_90_ ^a^ Positive ^b^		PRNT_90_ ^a^ Negative ^c^		*p*-Value
	F	%	f	%	
Degree of parenchyma volume reduction ^d^ (*n* = 81)					0.022
No reduction	3	20.0	12	80.0	
Mild/moderate	11	34.4	21	65.6	
Severe	20	58.8	14	41.2	
Brain calcifications ^d^ (*n* = 92)					0.591
No calcifications	1	16.7	5	83.3	
Subcortical	27	42.9	36	57.1	
Other sites	10	43.5	13	56.5	
Degree of ventriculomegaly ^d^ (*n* = 85)					0.136
No ventriculomegaly	3	25.0	9	75.0	
Mild/moderate	15	35.7	27	64.3	
Severe	17	54.8	14	45.2	
Type of ventriculomegaly ^d^ (*n* = 86)					0.556
No ventriculomegaly	3	25.0	9	75.0	
*Ex vacuo*	27	39.7	41	60.3	
Hypertensive	3	50.0	3	50.0	
Malformation of cortical development ^e^ (*n* = 90)					0.322
No	5	29.4	12	70.6	
Yes	31	42.5	42	57.5	
Cerebellum malformation ^e^ (*n* = 91)					0.739
No	29	40.9	42	59.1	
Yes	9	45.0	11	55.0	
Brainstem malformation ^e^ (*n* = 89)					0.311
No	29	39.2	45	60.8	
Yes	8	55.3	7	46.7	

^a^ 90% Plaque Reduction Neutralization Test. ^b^ Titer ≥ 10. ^c^ Titer < 10. ^d^ Fisher’s exact test. ^e^ Chi-square test.

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
