# Peer review of "Plaque Reduction Neutralization Test (PRNT) in the Congenital Zika Syndrome: Positivity and Associations with Laboratory, Clinical, and Imaging Characteristics"

_viruses, 2020, doi:10.3390/v12111244_

Round 1
Reviewer 1 Report
The manuscript by Ribeiro et al. is an interesting manuscript about congenital Zika syndrome and the plaque reduction neutralization test for ZIKV.
The authors study a cohort of 94 children born from 2005 to 2018 in the state of Maranhao (Brazil) with clinical or imaging characteristics of congenital zika syndrome. They analyze in details the CZS characteristics and assess the positivity of the plaque reduction neutralization test in children. They show that plaque reduction neutralization test is negative in more than 50% of children with CZS, suggesting they are false-negative. Importantly, the authors use a strain of ZIKV isolated in Brazil, hence the test’s negativity is unlikely due to the strain used. Thus, these results draw attention to the fact that the diagnosis of CZS should be primarily syndromic.
In conclusion, I recommend the publication of this manuscript in Viruses.
Minor changes:
- In the Materials and Methods, the authors have to add the exact name of ZIKV strain used for PRNT.
- In the title of tables, add “negative” in the sentence, as negative PRNT data are also shown:
“…. and positive and negative plaque reduction neutralization test…”.
Author Response
Response to Reviewer 1 Comments
Point 1: In the Materials and Methods, the authors have to add the exact name of ZIKV strain used for PRNT.
Response 1: The name of the ZIKV virus strain was added to the text 2.3. Plaque reduction neutralization test (PRNT90), line 126.
Thus, the ZIKV virus strain (ZIKV/H.sapiens/Brazil/PE243/2015-Asian) isolated in Brazil was used.
Point 2: In the title of tables, add “negative” in the sentence, as negative PRNT data are also shown: “…. and positive and negative plaque reduction neutralization test…”.
Response 2: “and negative” were added to the title of the five tables.
Lines 239 and 240: Table 1. Laboratory tests in children and mothers and positive and negative plaque reduction neutralization test in children. São Luís, Brazil, 2016-2018.
Lines 247 and 248: Table 2. Symptoms of infection in pregnant women and positive and negative plaque reduction neutralization test in children with CZS. São Luís, Brazil, 2016-2018.
Lines 253 and 254: Table 3. Clinical and cranial tomography characteristics of children with CZS and positive and negative plaque reduction neutralization test. São Luís, Brazil, 2016-2018.
Lines 257 to 259: Table 4. Means of the Z scores of the head circumference at birth and at 6, 12, 24 and 36 months and positive and negative plaque reduction neutralization test in cases of Congenital Zika Syndrome. São Luís, Brazil, 2016-2018.
Lines 264 and 265: Table 5. Changes in cranial tomography and positive and negative plaque reduction neutralization test in children with CZS. São Luís, Brazil, 2016-2018.
Appendix B
Lines 366 and 367: Table A1. Laboratory tests in children and mothers and positive and negative plaque reduction neutralization test in children. São Luís, Brazil, 2016-2018.
Lines 374 and 375: Table A2. Symptoms of infection in pregnant women and positive and negative plaque reduction neutralization test in children with CZS. São Luís, Brazil, 2016-2018.
Lines 379 and 380: Table A3. Clinical and cranial tomography characteristics of children with CZS and positive and negative plaque reduction neutralization test. São Luís, Brazil, 2016-2018.
Lines 385 to 387: Table A4. Means of the Z scores of the head circumference at birth and at 6, 12, 24 and 36 months and positive and negative plaque reduction neutralization test in cases of Congenital Zika Syndrome. São Luís, Brazil, 2016-2018.
Lines 391 and 392: Table A5. Changes in cranial tomography and positive and negative plaque reduction neutralization test in children with CZS. São Luís, Brazil, 2016-2018.
Reviewer 2 Report
The study by Ribeiro et al examines the case definition of Congenital Zika Syndrome and the various laboratory test results as well as clinical score and imaging results. Their analysis suggests that patients with CZS from other diagnostic measures, frequently have negative PRNT90 results and conclude that PRNT90 tests should not be used to diagnosis CZS.
Comments to improve the readability of the manuscript:
- Some of the introduction felt repetitive or choppy while reading. I would consider revising.
- The Materials and Methods section needs sub-headings so the reader can quickly find information on the ELISA kits or how head circumference was measured.
Author Response
Response to Reviewer 2 Comments
Point 1: Some of the introduction felt repetitive or choppy while reading. I would consider revising.
Response 1: The introduction was revised. Please see lines 68 to 100.
Confirmation of the Congenital Zika Syndrome (CZS) by specific laboratory tests became a challenge when the microcephaly epidemic started in Brazil in 2015 [1] and spread to other locations in Latin America [2,3]. The short duration of viremia [4-6], low blood viral loads [6], cross-reactions with other flaviviruses in serological tests [5,6], and difficulties in accessing specific and accurate diagnostic tests [1,5,7] have delayed or hampered laboratory confirmation of typical CZS clinical cases [1-3].
A positive ZIKV ribonucleic acid test (RNA NAT) in serum [5,7], urine [5], or cerebrospinal fluid (CSF) confirms CZS. A positive ZIKV IgM [5,7] in cases in which ZIKV RNA NAT was negative or was not performed should be considered a likely CZS case due to a possible cross-reaction with other flaviviruses (false-positive result). In this situation, a plaque reduction neutralization test (PRNT) positive for ZIKV and negative for dengue virus (DENV) is the confirmation criterion for CZS [5,9]. The CDC recommends that this test be run after 18 months of age when maternal antibodies will no longer be present in the child [5]. If both ZIKV RNA NAT and ZIKV IgM are negative, the child is unlikely to have CZS [5,8,9].
Difficulties in carrying out tests or inconclusive test results meant that most children born with congenital disabilities during and after the 2015 microcephaly outbreak were diagnosed with likely CZS [3,10-18]. A positive ZIKV IgM antibody by ELISA was the most widely used evidence for the presumed or confirmed diagnosis of CZS in several case series [10,12,13,15,18-26]. A negative ZIKV RT-PCR was found in some children with typical signs of CZS [13,15,17-19,21,22,25,27]. The ZIKV PRNT, a high-cost and time-consuming test [9], was hardly used for diagnosing CZS [20,24,26,28].
Although the CDC indicated ZIKV PRNT as a diagnostic criterion to confirm ZIKV infection and rule out cross-reactions (false-positive results) with DENV, three studies reported false negative results for this test. A study in a cohort of pregnant women with rash found low positivity (48.5%) of ZIKV PRNT in positive ZIKV RT-PCR pregnant women [29]. In a case-control study which included 91 cases of CZS with typical microcephaly (82 live births and nine stillbirths), 27 mothers had PRNT50 negative [22]. A study with 19 confirmed, possible or likely cases of CZS revealed negative PRNT in five normocephalic newborns, but with changes in cranial tomography, suggestive of CZS. Two of these newborns were born to ZIKV PRNT-positive mothers during pregnancy and had negative results for ZIKV IgM, syphilis, toxoplasmosis, cytomegalovirus, and rubella [20].
Based on possible false-negative PRNT results in children with likely CZS [20] and pregnant women with positive ZIKV RT-PCR [22,29], this study hypothesized that there are few differences in laboratory, clinical and imaging characteristics comparing positive and negative PRNT children with typical CZS.
Point 2: The Materials and Methods section needs sub-headings so the reader can quickly find information on the ELISA kits or how head circumference was measured.
Response 2: Sub-headings were added to the Materials and Methods. Please see lines 102, 110, 126, 142, 152, 198, and 205.
2.1. Type of study and data collection
2.2. Case definition
2.3. Plaque reduction neutralization test (PRNT90)
2.4. Anti-DENV and anti-ZIKV IgG antibodies
2.5. Clinical and imaging variables
2.6. Statistical Analysis
2.7. Ethics Statement